# Effect of pre-operative bicarbonate infusion on maternal and perinatal outcomes among women with obstructed labour in Mbale hospital: A double blind randomized controlled trial

Milton W. Musaba[1,2,3]*, Julius N. Wandabwa[1], Grace Ndeezi[3], Andrew D. Weeks[4], David Mukunya[4,5,6,7], Paul Waako[8], Victoria Nankabirwa[5,6], Kenneth Tulya-muhika Mugabe[2], Daniel Semakula[9], James K. Tumwine[3], Justus K. Barageine[10]

1 Department of Obstetrics and Gynaecology, Busitema University Faculty of Health Sciences, Mbale, Uganda, 2 Department of Obstetrics and Gynaecology, Mbale Regional Referral Hospital, Mbale, Uganda, 3 Department of Paediatrics and Child Health, School of Medicine, Makerere University College of Health Sciences, Kampala, Uganda, 4 Sanyu Research Unit, University of Liverpool/Liverpool Women's Hospital, Liverpool, United Kingdom, 5 Department of Epidemiology and Biostatistics, School of Public Health, Makerere University College of Health Sciences, Kampala, Uganda, 6 Centre for Intervention Science and Maternal Child health (CISMAC), Centre for International Health, University of Bergen, Bergen, Norway, 7 Department of Public and Community Health, Busitema University Faculty of Health Sciences, Mbale, Uganda, 8 Department of Pharmacology and Therapeutics, Busitema University Faculty of Health Sciences, Mbale, Uganda, 9 Africa Centre for Systematic Reviews and Knowledge Translation, Makerere University College of Health Sciences, Kampala, Uganda, 10 Department of Obstetrics & Gynaecology, School of Medicine, Makerere University College of Health Sciences, Kampala, Uganda

* miltonmusaba@gmail.com

## Abstract

### Introduction

Oral bicarbonate solution is known to improve both maternal and perinatal outcomes among women with abnormal labour (dystocia). Its effectiveness and safety among women with obstructed labour is not known.

### Objective

To determine the effect and safety of a single-dose preoperative infusion of sodium bicarbonate on maternal and fetal blood lactate and clinical outcomes among women with obstructed labour (OL) in Mbale hospital.

### Methods

We conducted a double blind, randomised controlled trial from July 2018 to September 2019. The participants were women with OL at term ($\geq$37 weeks gestation), carrying a singleton pregnancy with no other obstetric emergency, medical comorbidity or laboratory derangements.

**Data Availability Statement:** All relevant data are within the manuscript and its Supporting Information files.

**Funding:** Survival Pluss project grant number UGA-13-0030 at Makerere University supported this work through a PhD fellowship. Survival Pluss project is funded by The Norwegian Program for Capacity Development in Higher Education and Research for Development (NORHED) under The Norwegian Agency for Development Cooperation (NORAD).

**Competing interests:** The authors have declared that no competing interests exist.

### Intervention

A total of 477 women with OL were randomized to receive 50ml of 8.4% sodium bicarbonate (238 women) or 50 mL of 0.9% sodium chloride (239 women). In both the intervention and controls arms, each participant was preoperatively given a single dose intravenous bolus. Every participant received 1.5 L of normal saline in one hour as part of standard preoperative care.

### Outcome measures

Our primary outcome was the mean difference in maternal venous blood lactate at one hour between the two arms. The secondary outcomes were umbilical cord blood lactate levels at birth, neonatal sepsis and early neonatal death upto 7 days postnatal, as well as the side effects of sodium bicarbonate, primary postpartum hemorrhage, maternal sepsis and mortality at 14 days postpartum.

### Results

The median maternal venous lactate was 6.4 (IQR 3.3–12.3) in the intervention and 7.5 (IQR 4.0–15.8) in the control group, with a statistically non-significant median difference of 1.2 mmol/L; p-value = 0.087. Vargha and Delaney effect size was 0.46 (95% CI 0.40–0.51) implying very little if any effect at all.

### Conclusion

The 4.2g of preoperative intravenous sodium bicarbonate was safe but made little or no difference on blood lactate levels.

### Trial registration

PACTR201805003364421.

## Introduction

Obstructed labour (OL) is an important cause of maternal and perinatal morbidity and mortality in low resource settings [1,2]. In Uganda, OL directly accounts for 8% of all maternal deaths and up to 14.2% of the perinatal mortality rate [3–7]. In addition, OL indirectly contributes to 70% of all maternal deaths due to primary postpartum hemorrhage and sepsis [8,9].

Compared to normal labour, obstructed labour is associated with higher levels of lactic acid in maternal blood and amniotic fluid. Prolonged, strong and frequent uterine contractions impair blood supply/oxygenation to the placental bed which promotes anerobic respiration and the accumulation of its byproduct (lactate) [10,11]. Accumulation of lactic acid depletes the maternal and fetal buffer system with resultant metabolic acidosis. Towards the end of pregnancy, physiological alkalinisation of the myometrium is one of the compensatory mechanisms to ensure that the strength and efficiency of contractions is not compromised by the inevitable acidosis that occurs commonly in the second stage of normal labour [11–13]. Likewise, the fetus is also well prepared to regulate its own acid-base balance and counter the metabolic acidosis in several ways, provided the oxygen supply is adequate [14]. Unfortunately, when obstructed labour occurs and it is not relieved quickly, these compensatory mechanisms are depleted by the inadequate supply of oxygen [14]. This leads to accumulation of hydrogen

ions that easily cross the placental barrier to reach the fetus. These are known to cause low fetal pH, low Apgar scores and acidosis in the newborn [14]. Prolonged, uncorrected acidosis (accumulation of $H^+$ ions) leads to failure of basic cellular functions resulting into cellular death [10,11,13].

A potential way of improving maternal and perinatal outcomes among women with OL is to buffer the excess lactic acid, using a buffering agent as a form of intrapartum intrauterine resuscitation in the preoperative period while the women wait for emergency cesarean section [15,16]. Oral bicarbonate is a cheap and effective acid buffer, widely used in sports to improve performance [16,17]. It is not known to cross the placental barrier [18], but plays a key role in regulating the maternal and fetal acid-base chemistry [14,16]. Amongst women in dysfunctional labour, it probably improves fetal outcomes by altering the intrauterine conditions [16]. However, it is not known if bicarbonate is effective in improving maternal and perinatal outcomes among women with OL who are likely to be much more acidotic. In a recent trial, 4.26g of sodium bicarbonate dissolved in 200mls of drinking water was administered orally to women with abnormally progressing labour requiring augmentation and it was found to be effective in increasing the rate of spontaneous vaginal deliveries and reducing fetal acidosis without any adverse effects [16]. So, we chose to use a similar dose of 4.2 g (50mmol/L) of sodium bicarbonate 8.4% solution, administered parenterally as a single intravenous bolus dose since OL is an emergency that requires urgent intervention. The lactate was measured at the bedside using a hand held Lactate Pro2 (Arkray Factory, Shiga, Japan) point of care device with a high intraclass correlation coefficient [19].

Currently, intravenous fluid infusion (1.5L of normal saline) is a key component of the standard preoperative care for all women with OL [20,21]. Although this practice is probably adequate to correct the associated electrolyte imbalance and dehydration in most women, it may not be sufficient to completely reverse the associated metabolic acidosis. We hypothesized that 4.2 g (50 mmol/L) of sodium bicarbonate administered as a single intravenous dose in the preoperative period could reduce the maternal blood lactate at one hour after administration compared to normal saline alone. Our aim was to establish the effect and safety of a single-dose preoperative infusion of sodium bicarbonate on maternal and umbilical blood lactate levels and clinical outcomes among patients with OL.

## Materials and methods

### Study design

This was a double blind, randomized controlled trial. Preoperatively, half of the patients received an infusion of sodium bicarbonate (intervention arm) and the other half received no sodium bicarbonate (control arm).

### Study setting

The study was conducted in the labour ward of a regional referral and teaching hospital in eastern Uganda. Annually, approximately 12,000 births occur in this hospital and 35% of these have a caesarean section, 12% of these for OL. Most (> 90%) of the participants in this study attended at least one ANC visit, and more than two thirds were admitted as referrals from the lower health facilities in active labour [22].

### Participants

Over a period of 13 months, we screened and enrolled all eligible patients with OL, diagnosed by either an obstetrician or a Medical Officer using a definition of the American Association

of Obstetricians and Gynaecologists (ACOG). In the first stage of labour, she should have cervical dilatation >6 cm with ruptured membranes, adequate contractions lasting >4 hours with no change in cervical dilatation or delay in the second active stage of labour (nullipara >2 hours, multipara >1 hour) with adequate uterine contractions. In addition, any two of: the obvious signs of severe obstruction such as caput formation, severe moulding, Bandl's ring, subconjunctival haemorrhages or an oedematous vulva [23].

## Randomization

A sequence of random numbers was generated by an independent biostatician using an online randomization service at www.sealedenvelope.com in permuted block sizes of four, six and eight. An independent, off site pharmacist prepared and consecutively numbered identical study packages using the generated random numbers. Each package contained five similar 10 mL glass vials without the original labels. After consent for inclusion was confirmed, a study nurse would take the next numbered study drug package and administer its contents to the participant. In that way, both the participants and data collectors remained blind throughout the study.

## Intervention

Fifty milliliters of 8.4% sodium bicarbonate in identical 10 mL glass vials (Martindale Pharma, Essex, UK) administered intravenously as a single bolus dose (equivalent to 50 mmol/L, 4.2g).

## Comparator

Fifty milliliters of sodium chloride 0.9% in identical 10 mL glass vials (AccuHealth Care, Gujarat, India) administered intravenously as a single bolus dose.

In addition, all patients received the available Ministry of Health recommended standard preoperative care for all patients with OL which includes antibiotic prophylaxis, intravenous fluid replacement (1.5 L of normal saline), bladder drainage and lying in left-lateral position as they were being prepared for caesarean section [24].

## Measurements

The primary outcomes were maternal venous blood at one hour after administration of the study drug and neonatal umbilical artery blood lactate at birth, both measured at the bedside using a hand-held Lactate Pro2 device (Arkray Factory, Japan). The maternal blood lactate was measured both at enrollment and at one hour after administration of the study drug package. We further categorized the baseline maternal blood lactate levels into normal, preacidemia and acidemia using cut offs in the NICE guidelines for fetal scalp blood lactate because we could not find any published or widely used values of lactate levels among women with obstructed labour [25]. The secondary maternal outcomes of interest were; myometrial capillary blood lactate at cesarean section, primary postpartum hemorrhage (based on clinical diagnosis and treatment), puerperal sepsis, obstetric fistulae, and maternal death up to 14 days postpartum. The secondary perinatal outcomes were; neonatal umbilical vein blood lactate, Apgar score, admission to the neonatal care unit for any reason and perinatal death up to 7 days postnatal.

## Adverse reactions/events

These were derived from the package insert labelling for sodium bicarbonate and a review of the literature. The reactions consisted of increased thirst, continuing loss of appetite, swelling

of the lower limbs, venous irritation, muscle pain or twitching, continuing headache, nausea or vomiting, stomach crumps, unusual tiredness or weakness, nervousness or restlessness, mood or mental changes, frequent urge to urinate, slow breathing and cellulitis at the injection site. All reactions were monitored hourly for the first six hours, six hourly for the next 18 hours and then daily throughout the period of hospitalization.

## Sociodemographic, clinical and laboratory characteristics

Trained research assistants used an electronic interviewer administered questionnaire (Open Data Kit, ODK, www.getodk.org) to collect sociodemographic and clinical information from the participants at recruitment and; during their stay in the hospital up to 14 days postpartum. At baseline, five milliliters of maternal venous blood were collected; 2 mls in an Ethylenedi-aminetetraacetic acid (EDTA)vacutainer for a full blood count, and 3 mls in a general vacutainer (red top) for electrolytes, renal function tests, liver function tests and for grouping and cross matching. Ten milliliters of fresh urine were also collected in a sterile container for analysis. All the specimens were transported to MBN Clinical Laboratories in Mbale within one hour of collection for analysis. After administration of the study drug, all the participants were actively monitored by a research midwife up to 24 hours in order to identify any adverse effects.

## Sample size and power calculations

In order to detect a 15% difference in mean maternal venous blood lactate levels between the intervention and control groups, assuming a two-sided $\alpha$ of 5% and 90% power, 478 participants were required. This number was also adequate to detect a 15% difference in mean cord blood lactate between the intervention and control groups at birth. Further details are available in the published study protocol [23].

## Data collection and management

Six qualified midwives were trained as research assistants to collect data using a pretested interviewer administered electronic questionnaire on password protected smart phones running the ODK software. To increase accuracy, the data was supplemented by reviewing relevant health facility records such as the antenatal cards, the maternity and theatre registers and the participants' case notes. The questionnaire was coded with checks and skips to ensure internal consistency. Data on sociodemographic, clinical and laboratory parameters including maternal blood lactate was collected at baseline. Maternal blood lactate was measured again at one hour after study drug administration and newborn cord blood was tested for lactate levels at birth. The participants were followed for 7 days postnatal to ascertain the secondary perinatal outcomes and for 14 days postpartum to determine the secondary maternal outcomes. We included all deaths that occurred from the point of recruitment into the study, through delivery up to the 7[th] day postnatal (early neonatal deaths). In the first 24 hours after study drug administration, the participants were closely monitored to identify any drug reactions. After administration of the study drug, all the participants were actively monitored by a research midwife up to 24 hours. The women were examined and asked directly about the presence of each of the listed adverse/side effects. This was done hourly in the first six hours, followed by six hourly for the rest of the 18 hours and then daily throughout the period of hospitalization. To ensure data quality and completeness, the principal investigator would review the entries from the aggregate server every 24 hours.

## Statistical analysis

We used Stata software version 14.0 (StataCorp; College Station, TX, USA) and R statistical software version 3.6.0 (packages FSA, rcompanion, effsize, ggplot2 and ggpubr) for analyses. Our analysis included all participants who were enrolled and assigned to the two trial groups. We used the Shapiro-Wilks test and graphical methods to test for normality of the blood lactate measurements. We summarized categorical variables as frequencies and proportions and continuous variables as either means and standard deviations (SD) for parametric data or medians and interquartile ranges (IQR) presented as the 25th ($Q_1$) and 75th ($Q_3$) quartiles for non-parametric data. We assessed for associations between two categorical variables using the chi squared or Fischer's exact tests as appropriate [26]. For the primary outcomes, we compared the treatment groups using the Mann-Whitney U test and calculated effect sizes with corresponding 95% confidence intervals using methods described by Vargha and Delaney. The A measure is a value between 0 and 1: when the A measure is exactly 0.5, then the two treatments achieve equal performance; when A is less than 0.5, the active treatment is worse; and when A is more than 0.5, the control treatment is worse. The closer to 0.5, the smaller the difference between the two treatments; the further from 0.5, the larger the difference [27].

To determine the safety of sodium bicarbonate infusion we compared the incidence of side effects in each arm and report risk differences. We conducted prespecified subgroup analyses of the primary outcomes by referral status and by time from onset of study drug administration to delivery. This study was registered with the Pan African Clinical Trials Register (PACTR201805003364421) and the protocol was also published [23].

## Quality control

All six research assistants were qualified and experienced midwives. They were trained on the study protocol procedures during the trial dry run phase and at regular intervals throughout the implementation of the trial. They were also closely supervised by the PI. Our choice of the lactate Pro 2 device for measuring blood lactate was informed by the fact that it was maintenance free, accurate and did not require any calibration before and during use. MBN Clinical Laboratories Ltd are internationally accredited and involved in regular internal and external quality control checks. In order to ensure compliance and adherence with the standard operating procedures of the protocol, the PI would check all the case report and consent forms on submission to ensure completeness. The study site was monitored by a joint team from the School of Medicine Research and Ethics Committee (SOMREC), Uganda National Council for Science and Technology (UNCST) and the National Drug Authority (NDA).

## Ethical considerations

Ethical approval was obtained from the Makerere University School of Medicine Research and Ethics Committee (#REC REF 2017–103) and Uganda National Council for Science and Technology (HS217ES). We obtained written informed consent from each participant before admission to the study. The study received administrative clearance from Uganda National Drug Authority (NDA) and the Mbale Hospital research and ethics committee.

## Participant safety

All the serious adverse events (SAE's) were closely followed until resolution. They were reported to the School of Medicine Research and Ethics Committee (SOMREC) within 24 hours using standard reporting forms as detailed in the study protocol [23]. The data safety monitoring board (DSMB) met three times during the conduct of the trial to review the

progress of the study by reviewing the unblinded data, to ensure safety of the participants. The protocol had allowed for one DSMB meeting when a third of the patients had completed follow up, but the DSMB requested for another meeting at the half way mark in order to ascertain safety of the study participants.

### Patient public involvement

The patients and public were not involved in the design or conduct of this study.

## Results

### Study participants

From August 30, 2018 to September 15, 2019, there were 6,984 deliveries on the labour ward of Mbale Regional Referral Hospital. Of these,623 were diagnosed with OL. There were fewer births over the study period than previously reported by the hospital. This could have been due to inaccurate record keeping by the hospital or the series of industrial actions by the health workers, which disrupted service delivery in the public health sector. We excluded 146 women because they did not meet the eligibility criteria or declined participation, and so 477 were enrolled. Details of the reasons for exclusion of these women can be found in Fig 1. All the randomized women received the assigned treatments. One participant was diagnosed during surgery as having had a ruptured uterus with a fresh still birth, and this was reported to the ethical board as a protocol deviation. We obtained primary maternal and fetal outcomes for all participants. Thirteen women were lost to follow up after discharge from the hospital meaning that we could not ascertain all their secondary maternal and perinatal outcomes.

### Baseline characteristics

The baseline clinical and laboratory characteristics of the enrolled women were similar between the two groups. The mean age of the participants was 24 (±6) years, with a mean weight of 65 (±11) kg and height of 157 (±12) cm. At enrollment, the mean maternal hemoglobin level was12.2 (±1.9) mg/dL, with a median capillary blood lactate level of 6.9 (3.4–13) mmol/L. Participants in the intervention arm had a lower median lactate 6.6 (3.2–13.7) at baseline compared to those in the control arm 7.4 (3.7–12.4). Most of the participants 292/477 (61%) were acidotic (defined as maternal venous lactate >4.8 mmol/L) and the median duration from administration of study drug to childbirth) was 90 (50–148) minutes. Half of the participants 270/477 (57%) had used local herbs after the onset of labour and two-thirds 303/477 (64%) were admitted as referrals from the lower health facilities with a diagnosis of OL. The details of the baseline characteristics are in Table 1.

### Primary outcomes

The median maternal venous lactate at 1 hour was 6.4 (IQR 3.3,12.3) in the intervention and 7.5 (IQR 4.0,15.8) in the control group. The median difference of 1.2 mmol/Lin maternal venous blood lactate between the control and intervention groups was not statistically significant, p-value = 0.087. The Vergha and Delaney effect size (A) was 0.46 (95% CI 0.40,0.51) implying very little if any effect at all. Details of the outcomes are in Table 2.

### Secondary outcomes

Perinatal mortality was 81 per 1,000 live births in the intervention arm compared to 87 per 1,000 live births in the control arm, a risk ratio of 0.93, 95% CI (0.51,1.7). The risk of primary postpartum hemorrhage was 24/238 (10%) in the intervention arm compared to 29/239

Trial Flow Diagram

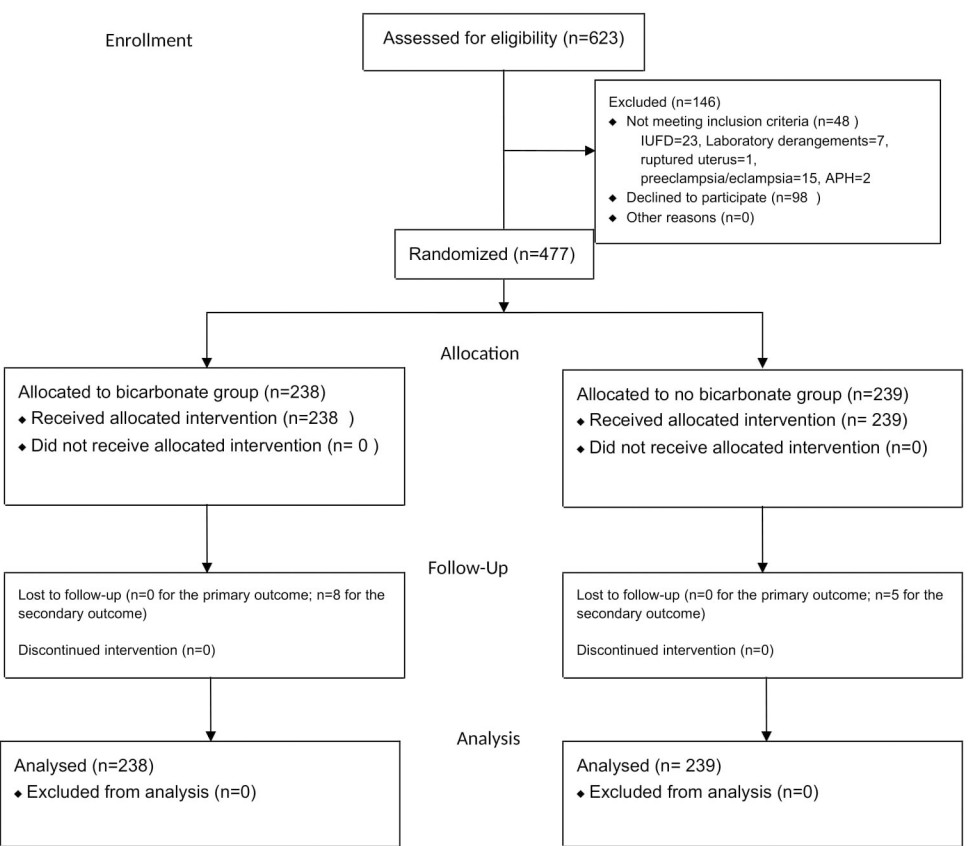

**Fig 1. Flow diagram for study participants in the sodium bicarbonate in obstructed labour trial.**

(12.2%) in the control arm, a risk ratio of 0.62, 95% CI (0.29,1.3). The risk of puerperal sepsis in the first 14 days after childbirth in the intervention arm was 4/238 (1.7%) compared to 11/239 (4.6%) in the control arm P = 0.072). Details of the rest of the secondary outcomes are in Table 3.

The risk of increased thirst was 44/238 (18.4%) in the intervention arm compared to 30/239 (12.6%) in the control arm, a risk ratio of 1.46, 95% CI (0.95,2.24). Two participants in the intervention group were diagnosed and treated for puerperal psychosis after being identified with mood and mental changes in the immediate postpartum period. The side effects and adverse drug reactions are shown in Table 4.

## Discussion

A single intravenous bolus infusion of 4.2g of sodium bicarbonate (equivalent to 50 mmol/L of bicarbonate) administered preoperatively to women with OL had very little if any effect at all on the maternal venous blood lactate level at one hour. Contrary to our prior belief, the administration of bicarbonate did not significantly reduce the maternal venous blood lactate at one hour, and the reduction in myometrial and fetal lactate levels were also not statistically significant. In the subgroup of women whose time from bicarbonate administration to birth was

**Table 1. Baseline characteristics of study participants, data presented as means (SD) or frequencies (%), except where specified.**

| Characteristic | All | Intervention | Control |
|---|---|---|---|
| | (N = 477) | (n = 238) | (n = 239) |
| Maternal age, years | 23.9 (6.0) | 23.1 (5.8) | 24.6 (6.1) |
| Maternal height, cm | 157.2 (11.8) | 157.4 (11.1) | 157.0 (12.5) |
| Maternal weight, kg | 64.6 (10.7) | 63.2 (9.8) | 66.1 (11.4) |
| Fetal heart rate on admission | 138.2 (14.1) | 138.4 (14.6) | 137.9 (13.6) |
| Gestational age in weeks | 38.6 (1.01) | 38.5 (1.0) | 38.6 (1.0) |
| **Maternal vital signs at recruitment** | | | |
| Maternal respiratory rate, bpm | 19.7 (3.2) | 19.9 (3.2) | 19.6 (3.2) |
| Maternal Pulse rate, bpm | 88.4 (15.0) | 88.4 (15.4) | 88.4 (14.7) |
| Maternal systolic blood Pressure, mmHg | 120.9 (11.0) | 120.5 (11.1) | 121.4 (10.9) |
| Maternal diastolic blood pressure, mmHg | 75.3 (10.6) | 74.8 (10.4) | 75.9 (10.8) |
| **Baseline maternal blood tests** | | | |
| Hemoglobin, mg/dL | 12.2 (1.9) | 12.1 (1.8) | 12.4 (1.9) |
| Drug administration to delivery interval, minutes* | 90 (50–148) | 90 (55–142) | 93.5 (46–155) |
| Less than 120 minutes | 295 (61.8) | 142 (59.7) | 153 (64.0) |
| More than 120 minutes | 182 (38.2) | 96 (40.3) | 86 (36.0) |
| Venous blood lactate, mmol/L* ($Q_1$ –$Q_3$) | 6.9 (3.4–13.0) | 6.6 (3.2–13.7) | 7.4 (3.7–12.4) |
| Normal < 4.2 | 149 (31.2) | 83 (34.9) | 66 (27.6) |
| Preacidemia (4.2–4.8) | 37 (7.8) | 17 (7.1) | 20 (8.4) |
| Acidemia > 4.8 | 291 (61.0) | 138 (58.0) | 153 (64.2) |
| Bicarbonate, mmol/L | 13.5 (5.7) | 13.5 (5.5) | 13.4 (5.9) |
| Sodium, mmol/L | 133.2 (5.1) | 133.3 (5.2) | 133.1 (5.0) |
| Potassium, mmol/L | 3.95 (0.72) | 3.9 (0.62) | 4.0 (0.81) |
| Chloride, mmol/L | 101.4 (4.95) | 101.5 (4.9) | 101.2 (5.0) |
| Calcium, mmol/L | 2.13 (0.26) | 2.13 (0.25) | 2.12 (0.27) |
| Magnesium, mmol/L | 0.73 (0.17) | 0.73 (0.19) | 0.72 (0.15) |
| Gravidity | | | |
| Primi-gravida | 260 (54.5) | 133 (55.9) | 127 (53.1) |
| Gravida 2 to 4 | 152 (31.9) | 76 (31.9) | 76 (31.8) |
| Gravida 5[+] | 65 (13.6) | 29 (12.2) | 36 (15.1) |
| Used local herbs labour** | | | |
| Yes | 270 (56.8) | 147 (51.5) | 123 (61.8) |
| Referred from another health facility | | | |
| Yes | 303 (63.5) | 160 (52.8) | 143 (47.2) |

* Medians with interquartile range (Q1 -Q3)

** Missing two observations (N = 475). There were no statistically significant differences between the two treatment groups.

more than 120 minutes however, the maternal myometrial lactate and fetal umbilical vein lactate were both significantly reduced.

The failure of the intervention to demonstrate a substantial effect could have been due to the high blood lactate levels in both arms, which remained high even after the intervention. The levels were more than twice the mean maternal venous lactate levels of 2.6±1 (± S.D.) mmol/L by the end of the first stage amongst 69 women without obstruction in a previous study [28]. Our choice to measure maternal lactate at one hour was informed by prior studies and the half-life of sodium bicarbonate, which is about 90 minutes [29,30]. The samples taken from the myometrium at CS and the umbilical cord could only be taken at the time of CS, and

**Table 2. Blood lactate results among women in the intervention and control groups.** Data presented as means (95% CI), N = 477.

| Outcome | Lactate: Median ($Q_1,Q_3$) | | Abs diff. in Medians | Effect size VDA** (95% CI) | P-value* |
|---|---|---|---|---|---|
| | Intervention (n = 238) | Control (n = 239) | | | |
| **Maternal venous blood lactate, mmol/L** | | | | | |
| At 1 hour | 6.4 (3.3,12.3) | 7.5 (4.0,15.9) | 1.2 | 0.46 (0.40,0.51) | 0.087 |
| Myometrial blood | 4.4 (2.9,6.8) | 4.8 (3.3,7.4) | 0.4 | 0.38 (0.33,0.44) | 0.077 |
| **Fetal umbilical blood lactate, mmol/L** | | | | | |
| Artery | 8.3 (5.3,13.85) | 8.8 (5.4,13.6) | 0.5 | 0.49 (0.44,0.55) | 0.775 |
| Vein | 7.4 (4.8,12.2) | 7.9 (5.0,12.2) | 0.6 | 0.48 (0.43,0.53) | 0.466 |

* Mann-Whitney U test

** VDA is Vargha and Dalaney's A, the significance tests of the A = 0.5; Abs diff. is Absolute difference.

were therefore often taken much later. The subgroup analysis (Table 5b in S2 File) explores this in more detail. There were significant differences between the groups in the myometrial lactate and umbilical vein lactate levels (which more closely reflects the myometrial levels following placental transfer). This was not however seen in the maternal samples taken at 1 hour after treatment, nor in those whose caesarean section was conducted at under120 minutes. This suggests that the bicarbonate and rehydration took more than one hour to have an effect on the lactate levels. With regard to the bicarbonate dose, we adopted a conservative regimen because of the limited information about its safety in a population of parturient women. It may be that a higher dose of bicarbonate, or the repeated maternal lactate assessments in future studies will show a delayed effect on myometrial and umbilical blood lactate levels.

Our findings of a greater reduction in maternal and arterial cord blood lactate among women diagnosed with OL in the facility and those that delivered within two hours of receiving the intervention suggests some evidence of a possible direct effect. Early intervention among those with less severe form of obstruction may be beneficial, (Table 5a and 5b in S2

**Table 3. Secondary clinical outcomes among women with obstructed labour.**

| Outcome | (Yes/No) | | | Risk difference | |
|---|---|---|---|---|---|
| | All (N = 477) | Intervention (n = 238) | Control (n = 239) | 95% CI | P-value |
| **Maternal** | | | | | |
| Primary postpartum hemorrhage | 53/424 | 29/209 | 24/215 | -0.02 (-0.78,0.35) | 0.275 |
| Puerperal sepsis | 15/462 | 11/227 | 4/235 | -0.03 (-0.06,0.001) | 0.072 |
| Ruptured uterus | 8/469 | 6/232 | 2/237 | -0.02 (-0.04,0.01) | 0.176 |
| Fistulae | 2/475 | 1/237 | 1/238 | - | 1.000 |
| Perineal tears | 8/468 | 3/235 | 5/234 | 0.01 (-0.01, 0.04) | 0.724 |
| Duration of hospital stay, days* | 3 (2–4) | 3 (2–4) | 3 (2–4) | - | 0.976 |
| Maternal respiratory rate at 1 hour, bpm** | 20 (8.4) | 20 (8.4) | 20 (8.4) | -0.11 (-0.64,0.42) | 0.686 |
| **Perinatal** | | | | | |
| Apgar Score < 7 | | | | | |
| At 1 minute | 118/359 | 59/179 | 59/180 | -0.001 (-0.08,0.08) | 0.532 |
| At 5 minutes | 42/435 | 22/216 | 20/219 | -0.01 (-0.06,0.04) | 0.750 |
| Admission to neonatal unit | 16/461 | 6/232 | 10/229 | 0.02 (-0.02,0.05) | 0.447 |
| Early neonatal death | 39/438 | 20/218 | 19/220 | -0.01 (-0.06,0.04) | 0.868 |

*Mann-Whitney test, Median (interquartile range).

** t-test, Mean (standard deviation).

**Table 4. Side effects of sodium bicarbonate infusion among women with obstructed labour.**

| Side effect | (Yes/No) | | | Risk difference | |
|---|---|---|---|---|---|
| | All (N = 477) | Intervention (n = 238) | Control (n = 239) | (95% CI) | P-value** |
| Increased thirst | 74/403 | 44/194 | 30/209 | 0.06 (-007,0.12) | 0.100 |
| Continuing loss of appetite | 15/462 | 9/229 | 6/233 | 0.01 (-0.02,0.04) | 0.601 |
| Swelling of the lower limbs | 19/458 | 11/227 | 8/231 | 0.01 (-0.02,0.05) | 0.641 |
| Muscle pain or twitching | 10/437 | 5/233 | 5/234 | 0 | 1.000 |
| Continuing headache | 16/461 | 11/227 | 5/234 | 0.03 (-0.01,0.06) | 0.202 |
| Nausea or vomiting | 11/466 | 7/231 | 4/235 | 0.01 (-0.01,0.04) | 0.544 |
| Stomach crumps | 5/477 | 3/235 | 2/237 | 0.00 (-0.01,0.02) | 1.000 |
| Unusual tiredness or weakness | 5/472 | 4/234 | 1/238 | 0.01 (-0.01,0.03) | 0.372 |
| Nervousness or restlessness | 3/474 | 1/237 | 2/237 | -0.004 (-0.02,0.01) | 0.623 |
| Mood or mental changes* | 2/475 | 2/236 | 0/239 | - | 0.499 |

* Both patients treated for puerperal psychosis

** Fischer's exact test.

File). In this study, 64% (303/477) of the participants were admitted as referrals with OL, a category which represents those with more severe prolonged and neglected obstruction in who the intervention was received late when severe complications had already occurred and could not be reversed. There were six perinatal deaths among women diagnosed with OL in the facility, and five of these were in the control arm (Table 5a in S2 File). So, further work may be needed to elucidate the importance of this result because it is possible that this represents a plausible true benefit that is time dependent.

Our results contrast findings from earlier studies [15,16]. In a recent trial from Sweden, a single oral dose of bicarbonate (4.26g dissolved in 200 mL of drinking water) improved the acid-base status, significantly lowered the amniotic fluid lactate levels and improved both the maternal and fetal outcomes among women with abnormally progressing labour undergoing augmentation with oxytocin (dystocia) [16]. This contrast in results could be attributed to the fact these two trials are not directly comparable because of differences in the study populations, routes of drug administration and strategies used to measure lactate. For instance, during labour, the lactate levels are higher in amniotic fluid compared to blood because it is locally produced by the fetus and myometrium [10,11,13]. Therefore, it is possible that the intervention could have significantly lowered the lactate levels in amniotic fluid which we did not measure. The ultimate goal in such an intervention would be to improve the intra uterine conditions, so we expected a parenteral dose of sodium bicarbonate to have a faster effect on the blood lactate levels compared to the oral route in an emergency situation. In this study we did not measure the amniotic fluid lactate levels, but in 92% (440/474) of the participants we measured the myometrial blood lactate at cesarean section and there was a substantial difference between the two groups. The median difference in myometrial blood lactate levels was more substantial in the sub-group of women that delivered at two or more hours after receiving the study drug (Table 5b in S2 File). This is the closest we could get to know what was happening in the uterine cavity.

We found no statistically significant differences in the incidence of selected secondary outcomes probably because the current study was not powered to study these clinical outcomes. Although high maternal lactate levels in the intrapartum period increase the risk of both maternal and fetal complications [31]. Hopefully, our findings will inform the design and conduct of future trials powered to detect differences in important clinical outcomes such as

puerperal and neonatal sepsis, primary postpartum hemorrhage (defined as blood loss > 500mls) and the duration of hospital stay.

Our finding of no evidence of difference in the occurrence of side effects/drug reactions between the two groups is in agreement with earlier reports regarding the good safety profile of this drug in laboring women. Further studies therefore, are still feasible [16,32].

The strength of this study is that it is a randomised controlled trial in which both the participants and the researchers were blinded. Blood lactate level was measured using a hand-held device at the bed side, which is suitable for a low resource setting without ready access to blood gas analyzers. To the best of our knowledge, this is the first trial from a low resource setting to determine the effect of bicarbonate on blood lactate and clinical outcomes among women with OL.

Our study had some limitations; our choice of a conservative dose regimen probably helped us avoid adverse drug reactions, but it may not have been effective in lowering the lactate levels for a sustained period of time. We did not measure the blood pH and acid-base status in this study because of logistical challenges and the existence of a huge body of knowledge that suggests lactate is a good proxy indicator for acidosis [32,33]. In addition, we used a cut off of 4.8 mmol/L for acidosis which has only been validated for fetal scalp monitoring because we could not find any other published and widely accepted cutoffs for maternal venous lactate in labour [25]. We were not powered to detect differences in important clinical outcomes.

## Conclusion

Our intervention of 4.2g of preoperative intravenous sodium bicarbonate was safe but made little or no difference to both the maternal and umbilical blood lactate levels. Our findings can be used as baseline data to inform further investigations in this area.

## Supporting information

**S1 Checklist. CONSORT checklist.**
(DOC)

**S1 File. Study protocol.**
(PDF)

**S2 File. Subgroup analysis.**
(DOCX)

**S1 Dataset. Anonymised data set.**
(DTA)

## Acknowledgments

We thank the study participants for accepting to be part of the study and the research midwives for working tirelessly to accomplish this task on time namely; Ms. Auma Prosscovia, Ms. Nandutu Sarah Waterah, Mrs. Atim Ketty Ojwar, Ms. Alibo Elizabeth, Ms. Sarah Talyewoya and Ms. Jessica Muduwa. Mr. John Mulangwa for handling the logistics and supply chain of the study drug. Finally, to Professor Thorkild Tylleskar for providing the electrochemical stripes for measuring lactate at the bedside.

## Author Contributions

**Conceptualization:** Milton W. Musaba, Julius N. Wandabwa, Grace Ndeezi, Andrew D. Weeks, Paul Waako, Victoria Nankabirwa, Kenneth Tulya-muhika Mugabe, James K. Tumwine, Justus K. Barageine.

**Data curation:** Milton W. Musaba, David Mukunya, Kenneth Tulya-muhika Mugabe, Daniel Semakula, Justus K. Barageine.

**Formal analysis:** Milton W. Musaba, Andrew D. Weeks, David Mukunya, Daniel Semakula.

**Funding acquisition:** Victoria Nankabirwa, James K. Tumwine.

**Investigation:** Milton W. Musaba, Julius N. Wandabwa, Grace Ndeezi, Andrew D. Weeks, Justus K. Barageine.

**Methodology:** Milton W. Musaba, Grace Ndeezi, Andrew D. Weeks, Paul Waako, Victoria Nankabirwa, Kenneth Tulya-muhika Mugabe, Daniel Semakula, James K. Tumwine, Justus K. Barageine.

**Project administration:** Milton W. Musaba, Julius N. Wandabwa, Grace Ndeezi, Paul Waako, Victoria Nankabirwa, James K. Tumwine.

**Resources:** Grace Ndeezi, Paul Waako, James K. Tumwine.

**Software:** Milton W. Musaba.

**Supervision:** Julius N. Wandabwa, Grace Ndeezi, Andrew D. Weeks, Justus K. Barageine.

**Visualization:** Justus K. Barageine.

**Writing – original draft:** Milton W. Musaba, Justus K. Barageine.

**Writing – review & editing:** Milton W. Musaba, Julius N. Wandabwa, Grace Ndeezi, Andrew D. Weeks, David Mukunya, Paul Waako, Victoria Nankabirwa, Kenneth Tulya-muhika Mugabe, Daniel Semakula, James K. Tumwine, Justus K. Barageine.

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
