## [Decision Letter · Decision Letter 0]

13 Nov 2020

PONE-D-20-25151

Effect of pre-operative bicarbonate infusion on maternal and perinatal outcomes among women with obstructed labour in Mbale hospital: a double blind randomized controlled trial.

PLOS ONE

Dear Dr. Musaba,

Thank you for submitting your manuscript to PLOS ONE. My apologies for the delaying in returning it to you. After careful consideration, we feel that it has merit but does not fully meet PLOS ONE’s publication criteria as it currently stands. Therefore, we invite you to submit a revised version of the manuscript that addresses the points raised during the review process.

Please consider the comments of both the reviewers. In addition I have the following comments:

1. Some of the reviewers requests for more details in the methods could be dealt with by referring the reader to the protocol. For example for the comment requesting further detail about the sample size calculation, it might be adequate to report in the relevant section of the text that further details are available in the protocol and to provide the reference.

2. It is commendable that a list of adverse effects was compiled and was actively monitored. However I agree with the reviewer that more details are needed about how data was collected for these. I think readers might want to know whether participants were asked directly for any of these effects and when?

3. In the methods section, bottom of page 6, after giving the protocol registration details I feel it would be useful to mention here that the protocol was also published.

4. I think it is adequate to state in the methods what the numbers in parenthesis for represent for means and medians but the authors might want to consider defining the interquartile range in the methods section as suggested.

5. With respect to the use of Vargha and Delaney A effect sizes, I feel it would be worth adding a sentence to the section on statistical analysis in the methods, on how these are interpreted. Readers might not be familiar with this less often used method of describing effect size. It might also be worth considering including this as a footnote to the Table. 

6. Would the authors consider whether or not they should report any statistical differences between the groups at baseline, perhaps in a footnote at the bottom of the table. I feel that some readers would look for these. I like the suggestion of adding BMI since it would give readers not familiar with the setting more idea about the nutritional status of the participants. Readers could be considering whether nutritional status could be a factor in determining outcomes. It might be more informative to give this as categorical data.    

7. I would like to see more about the interpretation of the blood lactate levels and how you determined that acidosis was present. I could only find this on Table 1. 

8. One reviewer commented on the need to thoroughly proof read this manuscript. While doing this please pay attention to spacing after a word and before a subsequent bracket, placement of commas immediately after a word and spacing after a semi-colon, particularly when used after IQR. 

9. Please ensure that readers are aware of the supplementary tables and that they are referred to where appropriate. 

10. There is some confusion about the perinatal outcome of death. I note that in the methods the mortality outcome stated is neonatal death but the result given is perinatal mortality. The denominator for perinatal mortality should be expressed in total births (and includes stillbirths). I agree with the reviewer that although the protocol excludes patients with intrauterine death (which I assume includes intrapartum death), I feel it would be important to report whether there were any intrapartum deaths after the diagnosis of OL. Perhaps this is the very outcome on which this intervention could have an effect. 

11. I feel that the conclusions in the abstract have a slightly different meaning than that state in the main text and perhaps the abstract states it better, (if it was clear that this included safety/adverse effects). We are moving away from statements about statistical significance towards statement about certainty of the effect. 

We look forward to receiving your revised manuscript.

Kind regards,

Jacqueline J. Ho, MB.ChB, MMedSc(ClinEpid), FRCP, FRCPCH, FRCPI

Academic Editor

PLOS ONE

Journal Requirements:

2. Please include a copy of Tables 5a and 5b which you refer to in your text on page 12.

3. Please include your tables as part of your main manuscript and remove the individual files. Please note that supplementary tables (should remain/ be uploaded) as separate "supporting information" files

Reviewers' comments:

Reviewer's Responses to Questions

**Comments to the Author**

1. Is the manuscript technically sound, and do the data support the conclusions?

Reviewer #1: Partly

Reviewer #2: Yes

2. Has the statistical analysis been performed appropriately and rigorously? 

Reviewer #1: Yes

Reviewer #2: Yes

3. Have the authors made all data underlying the findings in their manuscript fully available?

Reviewer #1: Yes

Reviewer #2: Yes

4. Is the manuscript presented in an intelligible fashion and written in standard English?

Reviewer #1: Yes

Reviewer #2: Yes

5. Review Comments to the Author

Reviewer #1: A two-arm randomized controlled clinical trial (n=477) was conducted to assess the effect and safety of sodium bicarbonate on maternal and fetal blood lactate and clinical outcomes in a cohort of women with obstructed labor.

Minor revisions:

1- Page 6: Sample size and power calculations section: Indicate the statistical testing method which achieved 90% power. If the testing method was a t-test, state the assumed standard deviation.

2- Page 7: Participant safety section: Indicate if AEs were collected according to a standardized method like CTCAE.

3- Page 8: Indicate the type of summary statistics provided in parenthesis. Specifically indicate the measure of variation for the median values.

4- Baseline characteristics: Include BMI in Table 1.

5- Table 1: When expressing IQR use first and third quartiles. Define IQR.

6- Draw conclusions that no differences were observed when the 95% CI for the risk ratios contain 1.0.

7- Thoroughly proofread the manuscript for grammatical errors.

8- Indicate the role of the funders.

Reviewer #2: Thank you very much for giving me the opportunity to read this important paper. I believe it is critical to research cost-effective methods to improve and perinatal outcomes related to obstructed labour in low-income settings. The methods are sound, and the manuscript is well-written. I have a couple of comments regarding this work.

1. In the first paragraph of the introduction you mention that: “obstructed labour accounts for 8% of all maternal deaths and up to 14.2% of the perinatal mortality rate”. In the next sentence you write that: “obstructed labour contributes to 70% of all maternal deaths due to primary postpartum hemorrhage and sepsis”. I can understand that obstructed labour can lead to sepsis and PPH, however not all women postpartum hemorrhage and all sepsis, have necessarily suffered from obstructed labour. In addition, I was not able to find a justification for the 70% in reference 8 and 9.

2. In the methods section under study setting you write that this regional teaching hospital has 12,000 births, 35% delivered by CS and 12% of whom have obstructed labour. This means that for a 13 month period approximately 1560 women with obstructed labour would have delivered. In the result section write that 623 women with obstructed labour were registered. You also write that “all eligible patients with obstructed labour” were screened and enrolled. Knowing the practical challenges with study inclusion especially in an acute situation and looking at the numbers mentioned above, might it be possible that not all patients with obstructed labour were screened and enrolled? Selection bias?

3. Would it be possible to elaborate a little more on the inclusion and exclusion criteria in the Methods section? At the same line, would it be possible to provide some more details in Figure 1, regarding on which grounds the 48 patients that were excluded?

4. In the result section, Table 1 describes the baseline characteristic. Would it be possible to add some more information about obstetric history (for example parity and previous caesarean section)? Did all women deliver by CS? And was there any difference between the interventions and control group for there characteristics?

5. Can you provide information on the time obstructed labour was diagnosed until the baby was delivered? Was there a difference between the intervention and the control group for this time?

6. You write in the results that there “was ruptured uterus with a fresh still birth.” Was this the only stillbirth in the study population? Were stillbirths excluded from the study (see also point 3)?

7. Table 3 shows the secondary outcomes. Under perinatal outcomes, neonatal deaths are mentioned. Neonatal deaths are normally defined as deaths within 4 weeks after delivery. Early-neonatal deaths are defined as deaths with 1 week after delivery. Stillbirths are those that die before the delivery and perinatal deaths are early neonatal deaths + stillbirths. Would it be possible to present both the early neonatal deaths and the stillbirths separately?

8. In the discussion you write “shown by our findings of a much higher reduction in maternal lactate among those that gave birth within two hours of receiving the intervention.” This statement should be substantiated with analysis presented in the results. I found the subgroup analysis in supporting information. There was no reference from the result section to the supplementary table, and results were not mentioned in the text either.

9. Would it be possible to reflect in the discussion on non-medical “side-effects” of the intervention such as additional costs and possible additional delay before a caesarean section is performed.

6. PLOS authors have the option to publish the peer review history of their article (what does this mean?). If published, this will include your full peer review and any attached files.

Reviewer #1: No

Reviewer #2: No

---

## [Author Response · Author response to Decision Letter 0]

10 Dec 2020

12th/12/2020

To 

Dr. Jacqueline J. Ho

Academic Editor

PLOS ONE

Dear Dr. Jacqueline J. Ho,

Re; Response to reviewers’ comments and resubmission of revised manuscript PONE-D-20-25151

Effect of pre-operative bicarbonate infusion on maternal and perinatal outcomes among women with obstructed labour in Mbale hospital: a double blind randomized controlled trial.

Thank you for reviewing and providing feedback on this manuscript. Please receive the revised copy with specific responses and changes summarized in the table below. 

Reviewers comment Response to comment Line number

Academic editor 

Please consider the comments of both the reviewers. In addition, I have the following comments: This has been done NA

1. Some of the reviewer’s requests for more details in the methods could be dealt with by referring the reader to the protocol. For example, for the comment requesting further detail about the sample size calculation, it might be adequate to report in the relevant section of the text that further details are available in the protocol and to provide the reference. Thank you for the guidance. We have updated the entire manuscript accordingly. Page 6

Lines 179 & 180

2. It is commendable that a list of adverse effects was compiled and was actively monitored. However, I agree with the reviewer that more details are needed about how data was collected for these. I think readers might want to know whether participants were asked directly for any of these effects and when? We have added a more detailed description under the data collection and management section.

“. After administration of the study drug, all the participants were actively monitored by a research midwife up to 24 hours. The women were examined and asked directly about the presence of side effects. This was done hourly in the first six hours, followed by six hourly for the rest of the 18 hours and then daily throughout the period of hospitalization. ” Page 6

Lines 193-196

3. In the methods section, bottom of page 6, after giving the protocol registration details, I feel it would be useful to mention here that the protocol was also published. We agree with you, this has been mentioned Page 7

Lines 219

4. I think it is adequate to state in the methods what the numbers in parenthesis for represent for means and medians but the authors might want to consider defining the interquartile range in the methods section as suggested. We have defined the interquartile range in the methods section under statistical analysis as the 25th (Q1) and 75th (Q3) quartiles.

We have also defined it in the footnotes of table 1 and 3. Page 7

Line 206

5. With respect to the use of Vargha and Delaney A effect sizes, I feel it would be worth adding a sentence to the section on statistical analysis in the methods, on how these are interpreted. Readers might not be familiar with this less often used method of describing effect size. It might also be worth considering including this as a footnote to the Table. We welcome this suggestion and this text has been added to the statistical analysis section

“The A measure is a value between 0 and 1: when the A measure is exactly 0.5, then the two treatments achieve equal performance; when A is less than 0.5, the first active treatment is worse; and when A is more than 0.5, the control treatment is worse. The closer to 0.5, the smaller the difference between the two treatments; the further from 0.5, the larger the difference” Page 7

Lines 210 - 214

6. Would the authors consider whether or not they should report any statistical differences between the groups at baseline, perhaps in a footnote at the bottom of the table. I feel that some readers would look for these. I like the suggestion of adding BMI since it would give readers not familiar with the setting more idea about the nutritional status of the participants. Readers could be considering whether nutritional status could be a factor in determining outcomes. It might be more informative to give this as categorical data. Thank you for the guidance, this foot note has been added to table 1 “There were no statistically significant differences between the two treatment groups”

This has been done, our decision to omit it was informed by the fact that a BMI obtained at the end of pregnancy would not be very informative. Page 9& 10

Lines 274 - 275

7. I would like to see more about the interpretation of the blood lactate levels and how you determined that acidosis was present. I could only find this on Table 1. Among women with obstructed labour, we could not find any published or widely used cut offs for blood lactate levels. So, we chose to use cut offs in the NICE guidelines for fetal scalp blood lactate. Page 17

Lines 404 - 407

8. One reviewer commented on the need to thoroughly proof read this manuscript. While doing this please pay attention to spacing after a word and before a subsequent bracket, placement of commas immediately after a word and spacing after a semi-colon, particularly when used after IQR. Thank you for this guidance. We have proof read the entire manuscript. We have also reviewed the text in detail and made several amendments to improve the grammar, flow of text and discussion. NA

9. Please ensure that readers are aware of the supplementary tables and that they are referred to where appropriate. This has been ensured by directing the readers to supplementary file 2 in the discussion section. Page 14 to 17

Lines 319 to 407 

10. There is some confusion about the perinatal outcome of death. I note that in the methods the mortality outcome stated is neonatal death but the result given is perinatal mortality. The denominator for perinatal mortality should be expressed in total births (and includes stillbirths). I agree with the reviewer that although the protocol excludes patients with intrauterine death (which I assume includes intrapartum death), I feel it would be important to report whether there were any intrapartum deaths after the diagnosis of OL. Perhaps this is the very outcome on which this intervention could have an effect. Sorry about the confusion, it should be perinatal death and not neonatal death because the reported number includes all deaths that occurred after recruitment into the trial up to the 7th day postnatal.

 All over the document

11. I feel that the conclusions in the abstract have a slightly different meaning than that state in the main text and perhaps the abstract states it better, (if it was clear that this included safety/adverse effects). We are moving away from statements about statistical significance towards statement about certainty of the effect. We have revised the conclusion in both the abstract and main text to make it uniform. Page 17

Line 408-414

Journal Requirements

1. Please ensure that your manuscript meets PLOS ONE's style requirements, including those for file naming. Thank you for this guidance. We have updated the entire manuscript and we believe it meets the journal requirements NA

2. Please include a copy of Tables 5a and 5b which you refer to in your text on page 12 This has been done Page 13 -14

Lines 312-317

3. Please include your tables as part of your main manuscript and remove the individual files. Please note that supplementary tables (should remain/ be uploaded) as separate "supporting information" file This is noted NA

Reviewer # 1:

Minor revisions:

1- Page 6: Sample size and power calculations section: Indicate the statistical testing method which achieved 90% power. If the testing method was a t-test, state the assumed standard deviation.

 This has been updated accordingly as advised by the academic editor. Page 6

Lines 176 & 180

2- Page 7: Participant safety section: Indicate if AEs were collected according to a standardized method like CTCAE. Yes, we used standard reporting forms provided by the School of Medicine Research and Ethics Committee (SOMREC). They are based on the CTCAE format. This has been added under the participant safety section. Page 8

Line 240

3- Page 8: Indicate the type of summary statistics provided in parenthesis. Specifically indicate the measure of variation for the median values This has been indicated as a foot note in all the tables Page 10

Line 275

4- Baseline characteristics: Include BMI in Table 1.

 This has been done, our decision to omit it was informed by the fact that a BMI obtained at the end of pregnancy would not be very informative. Page 8 and 9

Lines 274 275

5- Table 1: When expressing IQR use first and third quartiles. Define IQR.

 Thank you for this guidance. We have updated all the tables in the manuscript accordingly. Pages 9 – 14.

6- Draw conclusions that no differences were observed when the 95% CI for the risk ratios contain 1.0.

 Thank you for the observation, we have revised this accordingly NA

7- Thoroughly proofread the manuscript for grammatical errors.

 Thank you for this guidance. We have updated the entire manuscript and we believe it is better now. NA

8- Indicate the role of the funders.

 Survival Pluss Project at Makerere University. Funded by NORHED under NORAD. UGA-13-0030. The funders had no role in study design, data collection

and analysis, decision to publish, or preparation of the manuscript. Page 17

Lines 422 -425.

Reviewer #2:

Review summary 

Thank you very much for giving me the opportunity to read this important paper. I believe it is critical to research cost-effective methods to improve and perinatal outcomes related to obstructed labour in low-income settings. The methods are sound, and the manuscript is well-written. I have a couple of comments regarding this work.

1. In the first paragraph of the introduction you mention that: “obstructed labour accounts for 8% of all maternal deaths and up to 14.2% of the perinatal mortality rate”. In the next sentence you write that: “obstructed labour contributes to 70% of all maternal deaths due to primary postpartum hemorrhage and sepsis”. I can understand that obstructed labour can lead to sepsis and PPH, however not all women postpartum hemorrhage and all sepsis, have necessarily suffered from obstructed labour. In addition, I was not able to find a justification for the 70% in reference 8 and 9. We appreciate this comment, the paragraph has been revised accordingly to accurately reflect what we wanted to communicate. It was missing the words directly and indirectly.

 “In Uganda, OL directly accounts for 8% of all maternal deaths and up to 14.2% of the perinatal mortality rate. In addition, OL indirectly contributes to 70% of all maternal deaths due to primary postpartum hemorrhage and sepsis” Page 3

Lines 62 - 63

2. In the methods section under study setting you write that this regional teaching hospital has 12,000 births, 35% delivered by CS and 12% of whom have obstructed labour. This means that for a 13-month period approximately 1560 women with obstructed labour would have delivered. In the result section write that 623 women with obstructed labour were registered. You also write that “all eligible patients with obstructed labour” were screened and enrolled. Knowing the practical challenges with study inclusion especially in an acute situation and looking at the numbers mentioned above, might it be possible that not all patients with obstructed labour were screened and enrolled? Selection bias? The description in the study setting was based on previous hospital records, which may be inaccurate, although they include deliveries in the private wing of the hospital.

In the results section, we report the numbers that we received over the 13-month period of the study, excluding the deliveries in the private wing which are not usually many and obstructed labour is not a common occurrence. NA

3. Would it be possible to elaborate a little more on the inclusion and exclusion criteria in the Methods section? At the same line, would it be possible to provide some more details in Figure 1, regarding on which grounds the 48 patients that were excluded? Thank you we have elaborated more on this as advised by the academic editor under comment #3

Figure 1 has been revised accordingly, see supporting file 1. Page 8 and 20

Lines 255 - 519

4. In the result section, Table 1 describes the baseline characteristic. Would it be possible to add some more information about obstetric history (for example parity and previous caesarean section)? Did all women deliver by CS? And was there any difference between the interventions and control group for there characteristics?

 We have included the variable of parity as gravidity. Unfortunately, information on previous caesarean section was not collected but most of the patients were carrying the first pregnancy and 90% delivered by C/S. Page 9&10

Lines 274 - 276

5. Can you provide information on the time obstructed labour was diagnosed until the baby was delivered? Was there a difference between the intervention and the control group for this time? In table1, this variable is presented as “Drug administration to delivery interval, minutes”. Statistically the difference between the two times was not significant. Generally, the drug was administered shortly after diagnosing obstruction labour. Every effort was made to ensure that delivery was achieved within 120 minutes of the diagnosis. Page 9&10

Lines 274 - 276

6. You write in the results that there “was ruptured uterus with a fresh still birth.” Was this the only stillbirth in the study population? Were stillbirths excluded from the study (see also point 3)?

 Thank for this observation, this was the only patient with a ruptured uterus and stillbirth that was erroneously recruited into the trial and this diagnosis was made on the operating table. This was reported to the ethics board as a protocell deviation. Yes, stillbirths that happen before recruitment were excluded because having an intrauterine fetal death at recruitment was an exclusion criterion. We included all deaths that occurred from the point of recruitment into the study, through delivery up to the 7th day postnatal. Page 8

Lines 256 and 257

7. Table 3 shows the secondary outcomes. Under perinatal outcomes, neonatal deaths are mentioned. Neonatal deaths are normally defined as deaths within 4 weeks after delivery. Early-neonatal deaths are defined as deaths with 1 week after delivery. Stillbirths are those that die before the delivery and perinatal deaths are early neonatal deaths + stillbirths. Would it be possible to present both the early neonatal deaths and the stillbirths separately?

 This observation is very welcome.

We would like to refer to these as perinatal deaths and not neonatal deaths. This has been changed accordingly in table 3.

We included all deaths that occurred from the point of recruitment into the study, through delivery up to the 7th day postnatal. 

Unfortunately, it is not possible for us to present the early neonatal deaths and the stillbirths separately as request. Page 11&12

Lines 296 and 298

8. In the discussion you write “shown by our findings of a much higher reduction in maternal lactate among those that gave birth within two hours of receiving the intervention.” This statement should be substantiated with analysis presented in the results. I found the subgroup analysis in supporting information. There was no reference from the result section to the supplementary table, and results were not mentioned in the text either

 Thank you for the observation this has now been addressed accordingly by making some corrections in the discussion Pages 15Lines 339-349

9. Would it be possible to reflect in the discussion on non-medical “side-effects” of the intervention such as additional costs and possible additional delay before a caesarean section is performed

 We did not collect any information on the additional costs in this study.

Our anecdotal observations actually showed that the trial generally improved the quality of care that the patients received. We ensured that each patient the required an emergency CS did so in less than two hours of diagnosis by providing an essential surgical kit with all the necessary supplies for the surgery. Shortage of supplies in theatre is usually one of the common causes of delays. NA

END

---

## [Editor Report · Decision Letter 1]

14 Dec 2020

PONE-D-20-25151R1

Effect of pre-operative bicarbonate infusion on maternal and perinatal outcomes among women with obstructed labour in Mbale hospital: a double blind randomized controlled trial.

PLOS ONE

Dear Dr. Musaba,

Thank you for re-submitting your manuscript to PLOS ONE. We feel the revisions have improved it considerably. However there are still some minor revisions or clarifications that we feel are necessary before it fully meets PLOS ONE’s publication criteria. Therefore, we invite you to submit a revised version of the manuscript that addresses these additional points.

These additional points can be found on the attached document. This is the Response to Feedback form that you submitted. You will see below some of your responses I have added editors notes and these are where I would like you to further respond.  

We look forward to receiving your revised manuscript.

Kind regards,

Jacqueline J. Ho, MB.ChB, MMedSc(ClinEpid), FRCP, FRCPCH, FRCPI

Academic Editor

PLOS ONE

---

## [Author Response · Author response to Decision Letter 1]

17 Dec 2020

16th/12/2020

To 

Dr. Jacqueline J. Ho

Academic Editor

PLOS ONE

Dear Dr. Jacqueline J. Ho,

Re; Response to reviewers’ comments and resubmission of revised manuscript PONE-D-20-25151R1

Effect of pre-operative bicarbonate infusion on maternal and perinatal outcomes among women with obstructed labour in Mbale hospital: a double blind randomized controlled trial.

Thank you for reviewing and providing feedback on this manuscript. Please receive the revised copy with specific responses and changes summarized in the table below. 

Reviewers comment Response to comment Line number

Academic editor 

2. It is commendable that a list of adverse effects was compiled and was actively monitored. However, I agree with the reviewer that more details are needed about how data was collected for these. I think readers might want to know whether participants were asked directly for any of these effects and when? We have added a more detailed description under the data collection and management section.

“. After administration of the study drug, all the participants were actively monitored by a research midwife up to 24 hours. The women were examined and asked directly about the presence of each of the listed adverse/side effects. This was done hourly in the first six hours, followed by six hourly for the rest of the 18 hours and then daily throughout the period of hospitalization.” 

Editor: Were the women asked specifically about each of the listed adverse effects? 

Author: Yes. This has been updated accordingly. Page 6

Lines 197 and 198

6. Would the authors consider whether or not they should report any statistical differences between the groups at baseline, perhaps in a footnote at the bottom of the table. I feel that some readers would look for these. I like the suggestion of adding BMI since it would give readers not familiar with the setting more idea about the nutritional status of the participants. Readers could be considering whether nutritional status could be a factor in determining outcomes. It might be more informative to give this as categorical data. Thank you for the guidance, this foot note has been added to table 1 “There were no statistically significant differences between the two treatment groups”

This has been done, our decision to omit it was informed by the fact that a BMI obtained at the end of pregnancy would not be very informative.

Editor: Do you mean the included BMI's are taken at 

recruitment and not at booking? If so then I agree

with you that they are not appropriate and should

 not be included. I now wonder how many of these women had received antenatal care. Does this mean that the reader does not have sufficient information about the setting? Please consider whether you need to explain this. 

Author: All most everyone in this cohort attended ANC at least once ( 10.1371/journal.pone.0228856). This is in line with the UDHS survey findings. The main challenge was missing information on the ANC card where it was present. So, most of the height and weight measurements were done at enrollment.

This sentence has been added under the study setting section: “Most (> 90%) of the participants in this study attended at least one ANC visit, and more than two thirds were admitted as referrals from the lower health facilities in active labour.”

We have chosen to delete the BMI variable in table 1. Page 9& 10

Lines 274 – 275

Page 4 

Lines 115 -116

7. I would like to see more about the interpretation of the blood lactate levels and how you determined that acidosis was present. I could only find this on Table 1. Among women with obstructed labour, we could not find any published or widely used cut offs for blood lactate levels. So, we chose to use cut offs in the NICE guidelines for fetal scalp blood lactate.

Editor: Wouldn't it be better to have this in the methods under the description of the primary outcome but by all means if you consider it a limitation then is could be discussed again under the study limitations

Author: Agree. This has been added to the methods section under measurements as well. “We further categorized the baseline maternal blood lactate levels into normal, preacidemia and acidemia using cut offs in the NICE guidelines for fetal scalp blood lactate because we could not find any published or widely used values of lactate levels among women with obstructed labour.” Page 17

Lines 404 – 407

Page 5 

Lines 148 - 151

11. I feel that the conclusions in the abstract have a slightly different meaning than that state in the main text and perhaps the abstract states it better, (if it was clear that this included safety/adverse effects). We are moving away from statements about statistical significance towards statement about certainty of the effect. We have revised the conclusion in both the abstract and main text to make it uniform.

Editor: This better now. Rather than very ‘little difference’ I would like to suggest the wording ‘little or no difference’

Author: Agree. Thank you for the guidance Page 17

Line 408-414

Page 2 and 17

Lines 55- 56 and 399- 400

Journal Requirements

3. Please include your tables as part of your main manuscript and remove the individual files. Please note that supplementary tables (should remain/ be uploaded) as separate "supporting information" file This is noted

Editor: My apologies: We have given you conflicting advice since it was not understood that Table 5 was supporting information. Therefore, since you have now appropriately 

referred to the supporting information in the text Table 5 does not need to be included in the text

Author: Agree. Thank you for the guidance. We have removed table 5 from the text. NA

Reviewer #2:

1. In the first paragraph of the introduction you mention that: “obstructed labour accounts for 8% of all maternal deaths and up to 14.2% of the perinatal mortality rate”. In the next sentence you write that: “obstructed labour contributes to 70% of all maternal deaths due to primary postpartum hemorrhage and sepsis”. I can understand that obstructed labour can lead to sepsis and PPH, however not all women postpartum hemorrhage and all sepsis, have necessarily suffered from obstructed labour. In addition, I was not able to find a justification for the 70% in reference 8 and 9. We appreciate this comment, the paragraph has been revised accordingly to accurately reflect what we wanted to communicate. It was missing the words directly and indirectly.

 “In Uganda, OL directly accounts for 8% of all maternal deaths and up to 14.2% of the perinatal mortality rate. In addition, OL indirectly contributes to 70% of all maternal deaths due to primary postpartum hemorrhage and sepsis”

Editor: Agree. This is clearer now.

Author: Thank you Page 3

Lines 62 - 63

2. In the methods section under study setting you write that this regional teaching hospital has 12,000 births, 35% delivered by CS and 12% of whom have obstructed labour. This means that for a 13-month period approximately 1560 women with obstructed labour would have delivered. In the result section write that 623 women with obstructed labour were registered. You also write that “all eligible patients with obstructed labour” were screened and enrolled. Knowing the practical challenges with study inclusion especially in an acute situation and looking at the numbers mentioned above, might it be possible that not all patients with obstructed labour were screened and enrolled? Selection bias? The description in the study setting was based on previous hospital records, which may be inaccurate, although they include deliveries in the private wing of the hospital.

In the results section, we report the numbers that we received over the 13-month period of the study, excluding the deliveries in the private wing which are not usually many and obstructed labour is not a common occurrence.

Editor: It is possible that other readers could also be confused by this difference in birth and might attribute it to either poor reporting or poor recruitment by the authors. Do the authors need to discuss this difference? Perhaps at least that you can't explain why there were fewer births over the study period than previously reported by the hospital and that it could be due to inaccurate record keeping by the hospital. Or perhaps there are other possible explanations such as changes in the healthcare or referral system.

Author: Thank you for the guidance. We have added this in the results section under study participants: “There were fewer births over the study period than previously reported by the hospital. This could have been due to inaccurate record keeping by the hospital or the series of industrial actions by the health workers, which disrupted service delivery in the public health sector.” NA 

Page 8

Line 255-258

3. Would it be possible to elaborate a little more on the inclusion and exclusion criteria in the Methods section? At the same line, would it be possible to provide some more details in Figure 1, regarding on which grounds the 48 patients that were excluded? Thank you we have elaborated more on this as advised by the academic editor under comment #3

Figure 1 has been revised accordingly, see supporting file 1.

Editor: Rather than just refer the reader to Figure 1 I feel it would be clearer to make a statement such as: Details of the reasons for exclusion of these women can be found in Figure 1.

Author: Agree. Thank you for the guidance. We have now added this statement in the results section under study participants. Page 8 and 20

Lines 255 – 519

Page 8

Lines 260

7. Table 3 shows the secondary outcomes. Under perinatal outcomes, neonatal deaths are mentioned. Neonatal deaths are normally defined as deaths within 4 weeks after delivery. Early-neonatal deaths are defined as deaths with 1 week after delivery. Stillbirths are those that die before the delivery and perinatal deaths are early neonatal deaths + stillbirths. Would it be possible to present both the early neonatal deaths and the stillbirths separately?

 This observation is very welcome.

We would like to refer to these as perinatal deaths and not neonatal deaths. This has been changed accordingly in table 3.

We included all deaths that occurred from the point of recruitment into the study, through delivery up to the 7th day postnatal. 

Unfortunately, it is not possible for us to present the early neonatal deaths and the stillbirths separately as request.

Editor: Now that this is clearer, since perinatal death includes stillbirths and neonatal deaths in the first 7 days of life and the denominator for perinatal death is total births, I feel that this outcome is better described as ‘neonatal death occurring in the first 7 days of life’. We often refer to this as ‘Early neonatal death,’ (as did the reviewer), and this might be a better terminology for Table 3 provided that the description of outcomes in the methods uses this same terminology and defines early neonatal death (neonatal death in the first 7 days of life). 

I also see there is a superfluous 'yes and 'no' in the footnotes of Table 3 and Table 4. 

Author: Agree. Thank you for the guidance, we have updated this and defined it in the methods section. The superfluous yes and no in the footnotes of table 3 and 4 has been removed. Page 11&12

Lines 296 and 298

Page 6

Lines 192 and 194

9. Would it be possible to reflect in the discussion on non-medical “side-effects” of the intervention such as additional costs and possible additional delay before a caesarean section is performed

 We did not collect any information on the additional costs in this study.

Our anecdotal observations actually showed that the trial generally improved the quality of care that the patients received. We ensured that each patient the required an emergency CS did so in less than two hours of diagnosis by providing an essential surgical kit with all the necessary supplies for the surgery. Shortage of supplies in theatre is usually one of the common causes of delays. 

Editor: This might be interesting to the reader. You are suggesting that in the real world we might not expect to see CS occurring within 2 hours as frequently as in the trial. I wonder if this might have an impact on the study outcomes? Some of the subgroup analyses by duration since the intervention to birth suggested differences. 

Authors: Yes, unfortunately this is the reality. Based on findings from the national referral hospital which should ideally be better, 10.1186/s12884-020-03010-x. The median decision to delivery interval was 5.5 hrs. Our decision to do a subgroup analysis based on the time to delivery was informed by this fact. Yes, we think that it had an impact on the results. NA

END

---

## [Editor Report · Decision Letter 2]

22 Dec 2020

PONE-D-20-25151R2

Effect of pre-operative bicarbonate infusion on maternal and perinatal outcomes among women with obstructed labour in Mbale hospital: a double blind randomized controlled trial.

PLOS ONE

Dear Dr. Musaba,

Thank you for re-submitting your manuscript to PLOS ONE and thank you for responding to all the feedback. However there is just one minor issue in Table 1 that I suspect you intended to fix but overlooked. 

If you could correct as below and resubmit then I can accept the manuscript for publication.

The last two items on Table 1 are presented as Yes/No. "Used local herbs labour" and "Referred from another health facility". Please removed the 'No' line from the Table.

We look forward to receiving your revised manuscript.

Kind regards,

Jacqueline J. Ho, MB.ChB, MMedSc(ClinEpid), FRCP, FRCPCH, FRCPI

Academic Editor

PLOS ONE

---

## [Author Response · Author response to Decision Letter 2]

22 Dec 2020

22nd /12/2020

To 

Dr. Jacqueline J. Ho

Academic Editor

PLOS ONE

Dear Dr. Jacqueline J. Ho,

Re; Response to reviewers’ comments and resubmission of revised manuscript PONE-D-20-25151R2

Effect of pre-operative bicarbonate infusion on maternal and perinatal outcomes among women with obstructed labour in Mbale hospital: a double blind randomized controlled trial.

Thank you for reviewing and providing feedback on this manuscript. Please receive the revised copy with specific responses and changes summarized in the table below. 

Reviewers comment Response to comment Line number

Academic editor 

The last two items on Table 1 are presented as Yes/No. "Used local herbs labour" and "Referred from another health facility". Please removed the 'No' line from the Table. Agree. Thank you for the guidance. This has been corrected accordingly. Page 9 - 0

Lines 278 - 281

END

---

## [Editor Report · Decision Letter 3]

29 Dec 2020

PONE-D-20-25151R3

Effect of pre-operative bicarbonate infusion on maternal and perinatal outcomes among women with obstructed labour in Mbale hospital: a double blind randomized controlled trial.

PLOS ONE

Dear Dr. Musaba,

Thank you for revising Table 1 as requested. Do you think it would be better like this?

If you agree please revise and resubmit. 

A rebuttal letter that responds to each point raised by the academic editor and reviewer(s). You should upload this letter as a separate file labelled 'Response to Reviewers'.A marked-up copy of your manuscript that highlights changes made to the original version. You should upload this as a separate file labelled 'Revised Manuscript with Track Changes'.An unmarked version of your revised paper without tracked changes. You should upload this as a separate file labelled 'Manuscript'.

We look forward to receiving your revised manuscript.

Kind regards,

Jacqueline J. Ho, MB.ChB, MMedSc(ClinEpid), FRCP, FRCPCH, FRCPI

Academic Editor

PLOS ONE

---

## [Author Response · Author response to Decision Letter 3]

5 Jan 2021

29th /12/2020

To 

Dr. Jacqueline J. Ho

Academic Editor

PLOS ONE

Dear Dr. Jacqueline J. Ho,

Re; Response to reviewers’ comments and resubmission of revised manuscript PONE-D-20-25151R3

Effect of pre-operative bicarbonate infusion on maternal and perinatal outcomes among women with obstructed labour in Mbale hospital: a double blind randomized controlled trial.

Thank you for reviewing and providing feedback on this manuscript. Please receive the revised copy with specific responses and changes summarized in the table below. 

Reviewers comment Response to comment Line number

Academic editor 

Thank you for revising Table 1 as requested. Do you think it would be better like this? Yes, we agree. It looks alright Page 9 - 0

Lines 278 - 281

END

---

## [Editor Report · Decision Letter 4]

12 Jan 2021

Effect of pre-operative bicarbonate infusion on maternal and perinatal outcomes among women with obstructed labour in Mbale hospital: a double blind randomized controlled trial.

PONE-D-20-25151R4

Dear Dr. Musaba,

We’re pleased to inform you that your manuscript has been judged scientifically suitable for publication and will be formally accepted for publication once it meets all outstanding technical requirements.

Kind regards,

Jacqueline J. Ho, MB.ChB, MMedSc(ClinEpid), FRCP, FRCPCH, FRCPI

Academic Editor

PLOS ONE
---

## [Editor Report · Acceptance letter]

26 Jan 2021

PONE-D-20-25151R4 

Effect of pre-operative bicarbonate infusion on maternal and perinatal outcomes among women with obstructed labour in Mbale hospital: a double blind randomized controlled trial. 

Dear Dr. Musaba:

I'm pleased to inform you that your manuscript has been deemed suitable for publication in PLOS ONE. Congratulations! Your manuscript is now with our production department. 

Kind regards, 

on behalf of

Professor Jacqueline J. Ho 

Academic Editor

PLOS ONE